# Energy Saving with Zero Hot Spots: A Novel Power Control Approach for Sustainable and Stable Data Centers

Danyang Li [1], Yuqi Zhang [1], Jie Song [1,*], Hui Liu [2] and Jingqing Jiang [3]

1 Software College, Northeastern University, Shenyang 112000, China; 2110497@stu.neu.edu.cn (D.L.); 2190088@stu.neu.edu.cn (Y.Z.)
2 School of Metallurgy, Northeastern University, Shenyang 112000, China; liuhui@mail.neu.edu.cn
3 College of Computer Science and Technology, Inner Mongolia University for Nationalities, Tongliao 028000, China; jiangjingqing@imun.edu.cn
* Correspondence: songjie@mail.neu.edu.cn

**Abstract:** Data centers with high energy consumption have become a threat to urban sustainability on electric energy. In contrast, hot spots in a data center are another threat to server stability, which leads to unsafe data storage and service provisioning to urban lives. However, state-of-the-art works cannot ensure sustainability and stability together because they fail to consider them holistically. For example, some existing works eliminate the hot spots by increasing cooling power, which results in lower sustainability. In contrast, others reduce energy consumption by saving the cooling power, which harms stability. Therefore, to balance the hot spot elimination and energy saving through power control remains challenging, this paper proposes a novel power control approach for energy saving with zero hot spots in data centers. Power control works when hot spots appear, or consumed energy is excess. Specifically, we formulated a total consumption minimization problem to characterize and analyze the optimal set points for power control, where the number of hot spots is zero and the energy consumption is low. Adding the interactional penalty models can determine the power control approach when the objective function obtains the optimal solution. We propose a Modified Differential Evolution algorithm (MDE) to solve the function quickly and accurately. It adopts adaptive parameters to reduce the computing time. Meanwhile, it avoids optimal local solutions by changing mutation operations. Further, simulation experiments using our optimal solution demonstrate that energy consumption saves about 13% on average with zero hot spots, compared with three typical approaches.

**Keywords:** data center; hot spot elimination; energy consumption; differential evolution algorithm

## 1. Introduction

Data centers, providing a large number of services [1], are essential to our lives [2]. Their sustainability has a great impact on energy and the environment, while their stability is closely related to the safety of data storage and quality of service (QOS) [3]. Generally, the sustainability of data centers is known as their energy consumption capacity. In other words, the more energy a data center consumes, the less sustainable it becomes. Therefore, it is preferred that a data center consumes as little energy as possible. Correspondingly, the stability of data centers refers to the reliability of data storage and the ability to provide service, which is directly affected by the temperature of the rack in data centers [4]. In short, the more hotspots that exist in a data center, the less stable it is. Ideally, the number of hotspots should be zero.

In order to achieve sustainability and stability in data centers, much of the existing work has been devoted to two areas of research: One is the optimization of sustainability while ensuring that stability is within a certain threshold [5–7]; that is, optimizing system energy consumption without exceeding the temperature limit of the data center. The second is to fully protect the stability of the data center while ensuring that its sustainability is

within a certain threshold [8–10]. In other words, they optimize the system temperature within a reasonable energy consumption range. Both approaches are reasonable, but neither allows for a balance between sustainability and stability.

Almost all papers argue that energy saving and hotspot elimination are opposites, but we believe that the two cannot be seen in isolation; they are intrinsically linked. Furthermore, as the temperature of hotspots rises, the potential threat of degradation of service and other aspects of data centers from hotspots increases exponentially. Not only that, but the reduced compute rates generated by hotspots create greater computing capacity requirements, which in turn generates greater computing energy consumption, sending the data center into a vicious cycle. Obtaining a power control approach by weighing the intrinsic link between the computing power and cooling power, to ensure data center stability and sustainability is a huge challenge.

As previously mentioned, these studies eliminate hot spots, either by using constraints in their formulated cost minimization problems, or by setting broad metrics of hot spot elimination. In fact, it is necessary to consider hot spot elimination as an objective rather than a constraint for achieving cost minimization. This is clearly more advantageous as it can reduce heat accumulation quantitatively. In the long term, it improves not only the sustainability of the data center in terms of energy consumption, but also the stability in terms of QoS and data security. The higher QoS and data security are, the greater profit is. By contrast, it is the reduction in costs as a percentage of revenue, which can also be understood as reducing costs. Hence, saving energy and eliminating hot spots can jointly be implemented. We aim to provide a power control approach [11] that finds an energy cost-minimal operating point with zero hot spots and then supply methods for operating the data center, which considers the time and rate of hot spot elimination.

In this way, a power control approach for energy saving with zero hot spots is proposed. Our work adopts the task in a real data center, Ordos Uni Cloud Co., Ltd., (Ordos, China) [12], which is shown in Figure 1, to evaluate the performance of our approach.

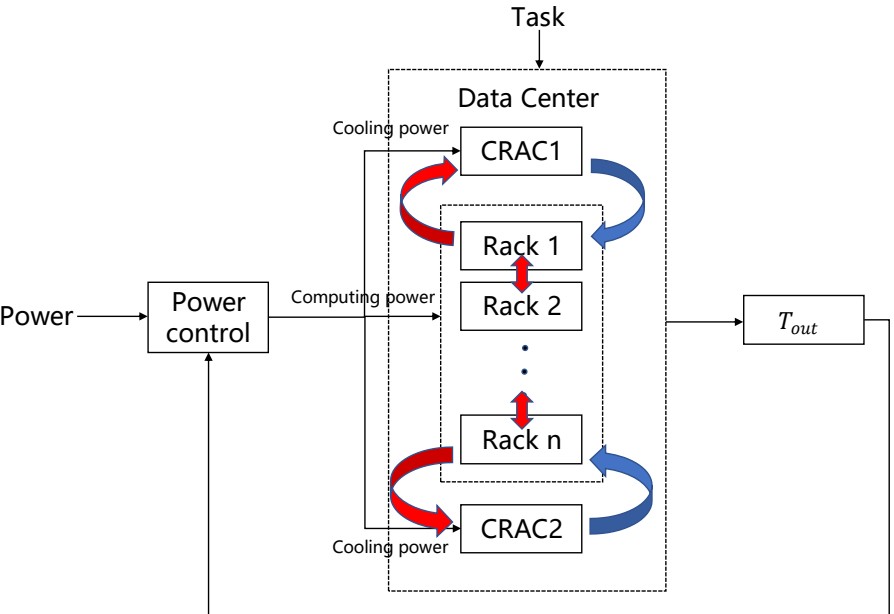

**Figure 1.** Power control architecture in data centers.

The contributions of this paper are twofold: First, from traditional methods to model power consumption in data centers and existing thermodynamical principles, we set up a model from which we derive an optimization problem that combines energy minimization with zero hot spots. In addition, we extended the minimal problem to incorporate the two-fold hot spot existence penalty formulas, which represent the numbers of hot spot existence and the time of hot spot elimination. Then, we limited the temperature constraint

and computing capacity constraint, which allows us to characterize the optimal solution better. Second, to solve the minimization problem, we propose a Modified Self-adjustive Differential Evolution algorithm (MSDE) to obtain a globally optimal solution that specifies the optimal power control for data centers in each time slot. In MSDE, the two parameters (*F* and *CR*) associated with the evolutionary process are updated to produce a flexible *DE* momently. However, the mutation operator adopts *DE*/current-to-gr-best/1 [10], which shows excellent performance in solving optimization problems.

The rest of this paper is organized as follows. Section 2 describes in detail the research content of the related work. Section 3 introduces the system model to be used in this paper. Sections 4 and 5 describe our control approach, which includes the objective function, constraints, and algorithm. Section 6 shows the simulation experiment of this paper and compares with four other methods. Finally, Section 7 concludes the paper.

## 2. Related Works

To achieve these goals, many researchers have made efforts in the last decade. Most papers [5–7] have been dedicated to maintaining the sustainability of the system, i.e., reducing the energy consumption of data centers, which can yield quick results and considerable economic benefits. In order to reduce energy consumption in data centers, some papers, such as [5], save on server start-up and shutdown costs by regulating tasks, which usually cause some task loss and task processing delays. Other papers, such as [6], reduce overall data center energy consumption by reducing cooling energy consumption, in which they usually place a constraint on the temperature in data centers. These loose constraints only keep the data centers away from the crash, but the resulting additional cooling energy consumption is not considered. The authors of [7] proposed a dynamic resource provisioning method, in which they predicted thermal effects to distribute power. However, as tasks change in real-time, so do their power requirements and corresponding cooling requirements. The accuracy of the prediction and the delay issue affect the effect of the approach. Although these methods save energy, they also generate stability threats.

Other papers [8–10] have focused more on the stability of the data centers, which take the number reduction in hot spots in data centers as the object of study. The TACS Thermal-Aware Control Strategy (TACS) [8] classifies tasks in a hierarchy and controls them separately. Although the heat distribution is balanced to a certain extent, its operation is complicated with a certain execution time delay. Authors in [9] present a Thermal-Aware Scheduling Algorithm (TASA), which assigns the hottest jobs to the coolest servers. However, the proposed scheduling does not take any remedial action when the threshold temperature is reached. In the last two years, papers have emerged that move towards a combination of sustainability and stability. For the first time, [10] took the stability-related factors, which are QOS issues caused by task arrival rates, as a part of the objective functions indicating total energy consumption, rather than only stability-related factors as constraints. However, it neglected to mention that excessive temperature over a long period of time, induced by the existence of hotspots, degrades stability and sustainability to a great extent.

Much research has been performed with the goal of saving energy while reducing hotspots. However, existing studies focus on energy saving with few hot spots, ignoring how hot spots are measured and how soon they can be eliminated [5]. These two overlooked points affect both the energy consumption and the number of hotspots, as shown in Figure 2. At this stage, the main problems in data centers are due to broad temperature constraints. The end result is a waste of energy for both computing and cooling. This is the problem that the approach of this paper aims to solve. Cooling a hotspot when it has been in place for some time requires more power and takes a longer time than cooling a hotspot when it is just becoming hot, resulting in more energy consumption. See the examples in Section 4.2 for detailed descriptions. Furthermore, setting the limit temperature for hotspots at the maximum value to guarantee safety only theoretically guarantees stable operation of the data center, which has little to no ability to cope with unexpected situations.

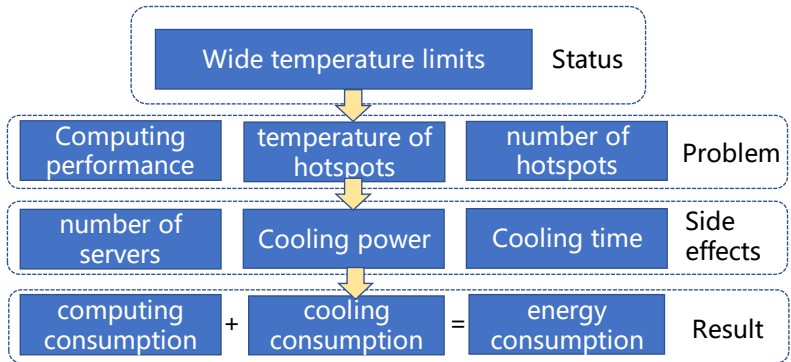

**Figure 2.** Existing problems in data centers.

In recent years, many studies have focused their approach to changing data center performance on the control of virtual machines (VMs). The authors of [13] priced the bandwidth between virtual machines to maximize network utility [14]. Automatic and efficient resource allocation for VMs using reinforcement learning [15], enables consolidated management of resources through migration of VMs.

Our work is to consider both the sustainability and the stability of the data center, i.e., to take into account the temperature and energy consumption of the data center and to provide an optimal power control method for the long-term operation of the data center.

## 3. System Model

In data centers, most of the equipment is identical, as this not only reduces maintenance complexity, but buying equipment in bulk reduces operating costs. In this case, the data center is considered to be homogeneous [16]. To put it more bluntly, the power and thermal models of all the servers and Computer Room Air Conditioners (CRACs) mentioned in this paper are the same. In this section, we list a range of models, including energy models and thermal models.

### 3.1. Computing Power Model

Previously, a number of papers have modeled the computing energy consumption in data centers [17]. They have constructed different models that are tailored to their respective research perspectives. As the focus of this paper is not on the details of the energy consumption model, a regression model, which is very popular and easy to understand, was chosen for this paper. In the regression model, the relationship between the power consumption of a server's functional units (CPU, memory, storage, etc.) and performance metrics is considered. The computing energy consumption of a server is characterized by capturing the fixed or idle power consumption as well as the variation in activity across the server's functional units.

$$P_{n,i}(t) = P_{idel} + (P_{active} - P_{idel}) \times n \tag{1}$$

where $P_{n,i}(t)$ is the power of the *n*-th server in the *i*-th rack at time *t*; $P_{idel}$ is the power when the server is idle; $P_{active}$ is the power when the server is working; and *n* is the number of the working servers.

### 3.2. Thermal Model

All electronic equipment must operate below its limiting temperature; otherwise, its lifecycle is shortened, and failure rates are higher, which affects the sustainability and stability of the data centers [13]. When a rack produces more heat, it may lead to a significant increase in the inlet temperature of other racks. Therefore, we must set the supply temperature of CRACs well-below the temperature limit to avoid inlet cooling air temperatures exceeding the threshold.

Since all the energy consumed by a rack is emitted in the form of heat, the racks in a data center can be considered as individual electric heaters when considering the heat model. In a data center, one side of the rack generates heat by using electricity, and on the other side, the CRAC also cools it by using electricity to generate cool air. It is well-known that the relationship between heat and temperature is as follows [6]:

$$Q(t) = \rho c_p f T(t) \tag{2}$$

where $\rho$ is the density of the air; $c_p$ is the specific heat capacity of the air; and $f$ is the air flow rate. So, according to the change in air temperature, we obtain a simplified thermodynamic equation for frame $i$:

$$m_i c_p \frac{d}{dt} T_{out}^i(t) = Q_{in}^i(t) - Q_{out}^i(t) + P_i(t) \tag{3}$$

where $T_{out}^i$ is the air temperature at the outlet of rack $i$; $m_i$ is the air mass in rack $i$; $Q_{in}^i(t)$ is the heat entering the rack; and $Q_{out}^i(t)$ is the heat output from the rack. If the cold air coming out of the CRAC can enter the rack without any influence at all, then the inlet air temperature of the rack would be directly equal to the output temperature of the CRAC. However, in reality, the air circulation within a data center is very complex, and the cold air output from the CRAC is influenced by the hot air generated by other racks. Therefore, the heat entering the rack can be expressed as:

$$Q_{in}^i(t) = \sum_{j=1}^{n} \gamma_{ji} O_{sup}^i(t) + O_{sup}^i(t) \tag{4}$$

where $Q_{sup}^i(t)$ is the heat supplied to rack $i$ by CRAC; $\gamma_{ji}$ is the percentage of flow from rack $j$ to rack $i$. The combined (3) and (4) has:

$$\frac{d}{dt} T_{out}^i(t) = \frac{\rho}{m_i} \left( \sum_{j=1}^{n} \gamma_{ji} T_{out}^j(t) - f_i T_{out}^i(t) \right) + \frac{\rho}{m_i} \left( f_i - \sum_{j=1}^{n} \gamma_{ji} f_i \right) T_{sup}(t) + \frac{1}{m_i c_p} P_i(t) \tag{5}$$

where $T_{sup}$ is the temperature supplied by CRAC; $f_i$ is the air flow rate in the rank $i$.

### 3.3. Cooling Power Model

Once the amount of heat dissipated by the data center racks is known, we can use the heat cycle between the computer racks to racks, and between the racks and the CRAC, to obtain the value of the heat that needs to be removed by the CRAC to find the power required. From Equation (6), we obtain the heat returned to CRAC from all racks.

$$Q_{ret}(t) = \rho c_p \sum_{j=1}^{n} \left( 1 - \sum_{j=1}^{n} \gamma_{ji} \right) f_i T_{out}^i(t) \tag{6}$$

Therefore, the heat sent to the ranks supplied by CRAC is $Q_{sup}(t) = \rho c_p f_{sup} T_{sup}(t)$. From this, the heat removed by CRAC is:

$$Q_{rem}(t) = Q_{ret}(t) - Q_{sup}(t) = \rho c_p \sum_{i=1}^{n} \left[ \left( 1 - \sum_{j=1}^{n} \gamma_{ji} \right) f_i \left( T_{out}^i(t) - T_{sup}(t) \right) \right] \tag{7}$$

To determine the amount of work that the CRAC must perform to remove a certain amount of heat, More et al. [6] introduced the coefficient of performance, $COP(T_{sup}(t))$, to indicate the efficiency of the CRAC as a function of the target supply temperature. The power consumption of the CRAC units can then be provided by:

$$P_{AC}\left( T_{out}(t), T_{sup}(t) \right) = \frac{Q_{rem}(t)}{COP(T_{sup}(t))} \tag{8}$$

where the *COP* [6] is the coefficient of performance of the CRAC set to supply cold air at Tsup temperature, and it characterizes the efficiency of a CRAC, i.e., it is defined as the

ratio of the amount of heat removed by the cooling device to the energy consumed by the cooling device performing the removal.

$$COP(T_{sup}(t)) = 0.0068T^2{}_{sup}(t) + 0.0008T_{sup}(t) + 0.458 \tag{9}$$

The cooling energy consumption of a data center depends largely on the amount of heat to be removed from it. As the amount of heat to be removed increases, the cooling energy required to remove it also increases, but this increase is not linear; as the heat accumulates, the power consumption required to remove it increases exponentially. Let us consider a small example to illustrate the influence of a small difference in supply temperature on the power consumption of the CRAC.

## 4. Problem Formulation

The objective of this paper is twofold: First, we plan to find optimal setpoints for the power distribution and supply temperature that minimize the power consumption of the data center. Therefore, from traditional methods to model power consumption in data centers and existing thermodynamical principles, we set up a model from which we derived an optimization problem that combines energy minimization. Second, we plan to ensure hot spot elimination. To this end, we extended the minimal problem to incorporate the hot spot existence penalty formulas, which represent the numbers of hot spot existence and the time of hot spot elimination. To ensure the QoS, we also added a task loss penalty to it. After this, we limited the temperature constraint and computing capacity constraint, which allows us to better characterize the optimal solution.

### 4.1. Total Cost Model

This section formulates an optimization problem for achieving an energy efficient and zero hotspot data center. Users send tasks to the data center through different types of applications, and as this phase is not the objective of this method of study, we directly translated the user demand ideally into the server utilization required to meet the user demand. The required computing power and the corresponding temperature variation were then calculated based on the user's server requirements. To ensure the appropriate data center temperature, we carried out the power regulation for cooling according to the solution of the optimization problem presented in this section. Through the power control method in this paper, we controlled the optimal computing power and cooling power at each time slot, while achieving data center hotspot elimination and energy savings. The objective function that combines energy saving and hot spot elimination is as follows:

$$C = C_{computing} + C_{cooling} + P_{hs} + P_e \tag{10}$$

$$C_{computing} = c \sum_{n=1}^{N} \sum_{i=1}^{I} \int_0^t P_i(t)dt \tag{11}$$

$$C_{cooling} = c \int_0^t P_{AC}dt \tag{12}$$

$$P_{hs} = \begin{cases} 0, \& T^i_{out} < T_0 \\ P_{hn} + P_{ht}, \ T^i_{out} \geq T_0 \end{cases} = \begin{cases} 0, \ T^i_{out} < T_0 \\ an + b(t_i - t_j), \ T^i_{out} \geq T_0 \end{cases} \tag{13}$$

$$P_e = e(P_i(t) + P_{AC} - \overline{P}) \tag{14}$$

where $C$ is the total cost of a data center in this paper; $e$ is the penalty parameter for energy overload; and $\overline{P}$ is the nominal parameter for average of power. $C_{computing}$ is the energy consumption of computing; $C_{cooling}$ is the energy consumption of cooling; $P_{hn}$ is the number penalty for hot spot presence; $P_{ht}$ is the time penalty for hot spot presence; $T_0$ is the limit temperature in data centers; $t_j$ is the time when hot spots disappear; $a$ is the penalty parameter for the number of hotspots appearance, which is related to total power; and $b$

is the penalty parameter for the time of hotspots presence. These two parameters directly determine the performance of our approach, and their specific determination method is obtained by multiple experimental comparisons.

*4.2. Constraint*

1.  Temperature constraint

The red line temperature varies from one design to another. Most of the red line temperatures are designed to be between 28 and 35 °C, which is not a problem in theoretical analysis, but in reality, the heat generation will be higher than the calculated value, and the heat dissipation will not reach the ideal condition at the time of the study, which may actually put the data center under a risky operation [17]. In this paper, the operating temperature is limited to 20–25 °C, leaving the system with a certain amount of cooling delay to allow for sustainable operation with zero hot spots.

$$20 \leq T_{out} \leq 25 \tag{15}$$

Let us consider a small example to illustrate the influence of a small difference in supply temperature on the power consumption of the CRAC. Consider the quadratic $COP(T_{sup})$ and two cases where the returned air must be cooled down to 20 °C, in the first case from 25 to 20 °C and in the second case from 35 to 20 °C. Normally, the density of air in a data center is 1.3 kg/m$^3$ and the specific heat capacity of air is 1 kj/kg·c, where the air flow rate when the fan is working is about 10 m/s. From Equation (2), it can be obtained that in the first case, the heat to be removed by dropping 5 °C is 65 W, and in the second case, the heat to be removed by dropping 15 °C is 195 W. From Equation (9), $COP$ (20) = 3.19. We can assume that it takes 1 min to lower 1 °C. By Equation (12), the energy consumed by the CRAC to cool down the returned air to the required temperature is $P_{AC,1} = \frac{65}{3.19} \times 300 = 6114$ W, and $P_{AC,2} = \frac{195}{3.19} \times 450 = 55,008$ W.

Here, it is shown that if the upper temperature rises by 10 °C, lowering it to the same temperature requires nine times more energy.

2.  Power input constraint

Power input constraint means that the input power requirement for server operation is less than the maximum power that can be provided by the data center power supply.

$$P_i(t) \leq P_{max} \tag{16}$$

3.  The total power demand constraint

The total power demand constraint, i.e., the computing capacity constraint, is the total energy consumption of all running servers that cannot exceed the maximum energy required for computing in the data center.

$$P_{total} \leq P_d \tag{17}$$

where $P_d$ is the upper bound of total power demand.

## 5. Modified Self-Adaptive Differential Evolution Algorithm

It is worthless that the objective function in this paper is a constrained, non-linear optimization problem; the constraint equation is complex. In recent years, evolutionary algorithms have been used by experts and scholars in solving optimization problems. Evolutionary algorithms are biomimetic algorithms constructed by simulating the biological evolutionary process of natural selection and genetics of Darwinian biological evolution. It can search for the optimal solution to the evaluation function by simulating the iterative process of natural evolution. Among them, the differential evolution algorithm is suitable for solving real-number optimization problems and has been successfully applied to various real-world optimization problems [18]. The algorithm has been used extensively in practical

applications and has proved to converge well. However, it is not an easy task to set the control parameters correctly when solving practical problems. This paper is therefore inspired by the adaptive differential evolution algorithm and uses adaptive parameter settings in solving the optimization problem in this paper. However, although this can provide the optimal solution while the parameters are adaptive, it may fall into a local optimum. For this reason, we further improve the variational step of the parameter adaptive differential evolution algorithm by proposing an improved differential evolution algorithm that eliminates the drawback of unselectable parameters and avoids falling into a local optimum. The final power allocation method is determined based on the results obtained from the Modified Self-Adjustive Differential Evolution Algorithm (MSDE) to ensure that the data center can achieve minimum energy consumption while having zero hotspots. The flow chart of the MSDE is as follows in the Figure 3.

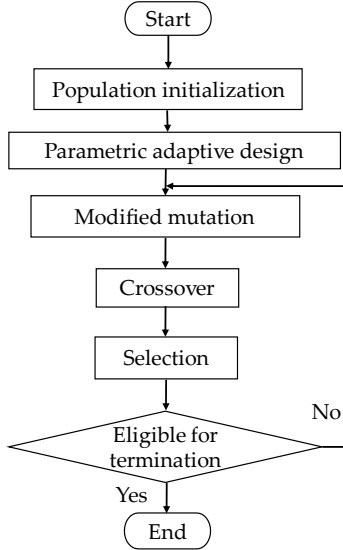

**Figure 3.** The flow chart of the MSDE.

### 5.1. Individual Encoding Structure

In Equation (10), this work aims to specify the optimal power control approach among temperatures for data centers in each time slot $\tau$. Each individual $i$ in MDE is the cooling power. Then, it is encoded as follows:

$$x_i = [P_{cooling}^{\tau,1}, \ P_{cooling}^{\tau,2} \cdots \cdots P_{cooling}^{\tau,N}] \tag{18}$$

In this way, the approach of transferring between the population and the solution of $P$ is obtained.

### 5.2. Population Initialization

The population is randomly initialized according to a uniform distribution within the feasible search space of decision variables; $x_0$ denotes the value of decision variable $j$ of $i$, $j$ individual $i$ ($i \in \{1, 2,..., \chi\}$) in the first generation, and $\chi$ denotes the size of the population. The population is initialized as follows:

$$x_i = x_i^{min} + r_i(x_i^{max} - x_i^{min}) \tag{19}$$

### 5.3. Parametric Adaptive Design

Choosing suitable control parameter values is, frequently, a problem-dependent task. The trial-and-error method used for tuning the control parameters requires multiple optimization runs. In this section, we propose a self-adaptive approach for control parameters. Each individual in the population is extended with parameter values. In Figure 1, the

control parameters that will be adjusted by means of evolution are *F* and *CR*. Both of them are applied at the individual level. The better values of these control parameters lead to better individuals which, in turn, are more likely to survive and produce offspring and propagate these better parameter values.

$$F_{i,G+1} = \begin{cases} F_l + rand_1 F_u, & if \ rand_2 < \tau_1 \\ F_{i,G} \ , & otherwise \end{cases} \tag{20}$$

$$CR_{i,G} = \begin{cases} rand_3, & if \ rand_4 < \tau_2 \\ CR_{i,G}, & otherwise \end{cases} \tag{21}$$

They produce factors *F* and *CR* in a new parent vector. $Rand_j$, $j \in \{1, 2, 3, 4\}$, are uniform random values. After many experiments, we set $\tau_1 = \tau_2 = 0.1$. Based on $F_l = 0.1$, $F_u = 0.9$, the new *F* takes a value from (0.1, 1.0) in a random manner. The new *CR* takes a value from (0, 1). $F_{i,G+1}$ and $CR_{i,G}$ are obtained before the mutation is performed. So they influence the mutation, crossover, and selection operations of the new vector $X_{i,G+1}$.

### 5.4. Mutation-DE/Current-to-gr-Best/1

The oldest of the DE [19] mutation schemes is DE/rand/1/bin, developed by Storn and Price [12], which is said to be the most successful and widely used scheme in the literature [20]. However, references [21,22] indicate that DE/best/2 and DE/best/1 may have some advantages over DE/rand/1. The authors of [23] argue that it is beneficial to merge information about the best solution (with the lowest objective function value of the minimization problem) and use DE/current-to-best/1 in their algorithm. Greedy strategies [24] such as DE/current-to-best/k and DE/best/k, compared with DE/rand/k, lead to the best solution by leading an evolutionary search to the best solution found so far, thus converging faster to that point, and thus benefiting from their fast convergence. However, due to this tendency to exploit, in many cases populations may lose their diversity and global exploration ability within a relatively small number of generations, thus falling into some local optimum point in the search space. Furthermore, DE employs a greedy selection strategy (choosing the better one between the target vector and the trial vector), using a fixed scale factor *F*.

$$\vec{V}_{i,G} = \vec{X}_{i,G} + F_i(\vec{X}_{gr-best,G} - \vec{X}_{i,G} + \vec{X}_{r_1^i,G} - \vec{X}_{r_2^i,G}) \tag{22}$$

### 5.5. Crossover and Selection

Following the algorithmic flow of the differential evolution algorithm (Algorithm 1), a crossover operation is performed after the mutation. The crossover is executed based on Equation (23), where the values of the crossed individuals are randomly selected from the corresponding values of the variant individuals or the corresponding original individuals. The generated random number is compared with the crossover probability to decide whether the crossover operation is executed or not, and the crossover individuals are thus obtained, as shown in Equation (23):

$$x_i''(g) = \begin{cases} x_i(g) & if \ rand \ (0,1) < CR \\ x_i(g) & else \end{cases} \tag{23}$$

Theoretically, a good population should satisfy both convergence and diversity to a certain extent to avoid premature convergence or search for a local minimum. Thus, in practice, researchers usually use strategies such as roulette or tournament selection when selecting parents to ensure that the parents are good while ensuring a certain diversity among parent vectors to increase the likelihood of producing good individuals in future generations. In the subsequent selection operation in the differential evolution algorithm, the newly generated individuals are selected by greedy rules, and the better individuals

are selected by an elite retention strategy to build a new generation of high-performance populations, as shown in Equation (24).

$$x_i(g+1) = \begin{cases} x_i''(g), & f(x_i''(g)) < f(x_i(g)) \\ x_i(g), & else \end{cases} \tag{24}$$

---

**Algorithm 1** HE: Hot spot elimination and Energy saving.

---

**Input**:

Task load to be computed in the observed time.

1. **Begin**
2. Parameter settings
3. $a, b, c, e, n, i, \gamma_{ji}, m, c_p, \rho, P_{active}, P_{idle}, P_{max}, P_d, \overline{P}$ ← BasicParameterSet()
4. $J, T_{0,sup}$, ← InterParameterSet()
5. Perform the Initialization with (19)
6. Parametric adaptive design with (20) and (21)
7. $g \leftarrow 1$
8. **While** $g \leq G$ do
9. **For i** ← 1 to I do
10. Perform the mutation with(22)
11. Perform the crossover with (23)
12. Perform the selection with (24)
13. **End for**
14. $g \leftarrow g + 1$
15. **End while**

**Output:**

Solution $S$ contains supply temperature $T_{sup}$, computing power $P_{computing}$ and minimized total cost $C_{min}$.

---

## 6. Performance Evaluation

In this section, extensive simulations are conducted to evaluate the effectiveness of the approach proposed in this paper. The performance of this method is compared with other typical common methods [25] in terms of computing power consumption, cooling power consumption, the number of hotspots, energy consumption, and total cost. It is assumed that the data center is homogeneous, and that the electricity cost is constant. We use workload traces from a real data center of Erdos UniCloud Ltd., Inner Mongolia, China [26].

### 6.1. Parameters Setting

Our simulated data center consists of 10 homogeneous server racks, i.e., all 10 racks have the same power characteristics, safety temperature thresholds, and physical parameters. The rack model is a Dell PowerEdge 1855 with 10 single processor blade servers, i.e., a total of 10 CPU units per rack. The processor power consumption is shown in Figure 4. In general, the safety threshold temperature is 30 °C. To ensure that it can always operate under the zero hot spot requirement of this paper, we set the safety threshold temperature in the data center to 25 °C. The load trace we provided to the data center includes, for Erdos UniCloud Ltd., a request for arrival records for four real-world scenarios over the course of a week with a time density of 1 h, as shown in Figure 5. In order to verify the effectiveness and generality of our approach, four different cases of app requests were selected as tasks for the data center. For ease of calculation, we do not take into account the peak-to-valley difference in electricity consumption. All are calculated at RMB 0.5/kWh for ideal conditions.

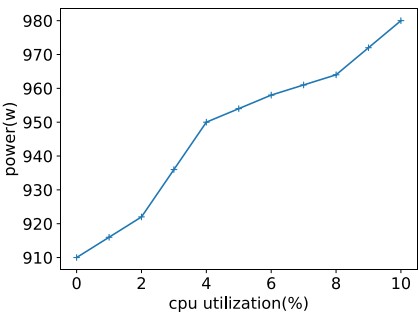

**Figure 4.** Power consumption for a single server.

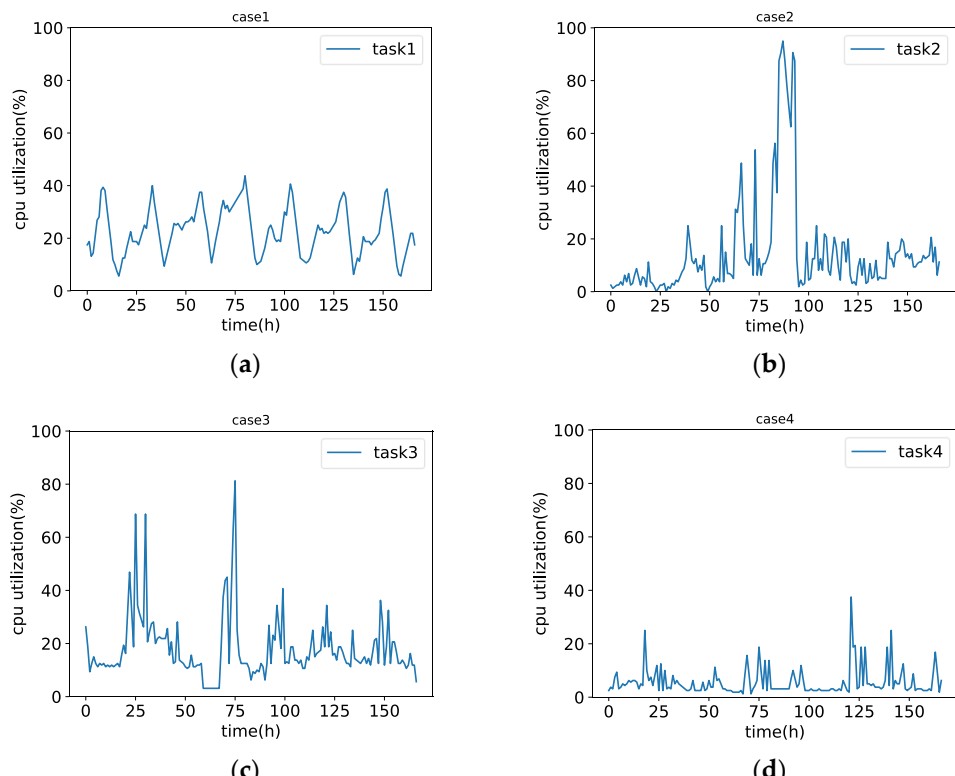

**Figure 5.** Workload for 4 cases: (**a**) case 1, (**b**) case 2, (**c**) case 3, (**d**) case 4.

## 6.2. Experimental Results

Figure 6 shows the power control approach for different load cases. It shows the control method for computing power consumption and cooling power consumption for a time density of 1 h during a week. It can be noticed that as the load increases, the computational power consumption increases, and thus the heat caused by the computation increases, creating a tendency for the rack exit temperature to increase. As the outlet temperature increases, the cooling equipment requires an increase in cooling power consumption as a means of reducing the rack temperature and keeping it under the limit at all times. In more detail, from the comparison of the four cases we can also find that the cooling power is lower than the computing power when the task demand is less than 50%, and the cooling power response is slower due to the low computing power at this time, as in Figure 6a,d.

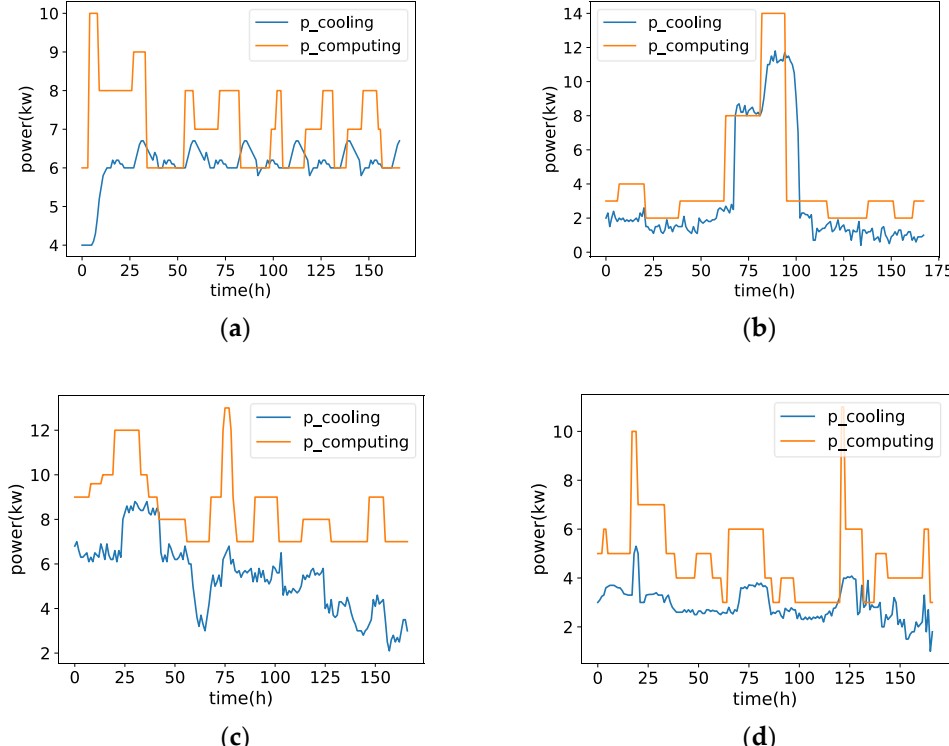

**Figure 6.** The results of power control approaches (**a**) case 1, (**b**) case 2, (**c**) case 3, (**d**) case 4.

### 6.3. Comparison Results

To demonstrate the performance of our work, we first compared it with several state-of-the-art control approaches for sustainable and stainable data centers, including First Come First Serve (FCFS), Thermal-Aware Scheduling Algorithm (TASA) [17], and thermal-aware control strategy (TACS) [5].

FCFS is possibly the most straightforward scheduling approach. The jobs are submitted to the scheduler, which dispatches the jobs based on the order of the jobs received

TASA is based on the theory of the coolest inlet that performs the assignment of the hottest jobs to the coolest servers. The TASA sorts the servers in the increasing order of the temperatures. The jobs are sorted in a similar way but in the reverse order, such that the hottest job is first in the order. The hottest job is assigned to the coolest server, and the thermal map of all the servers is updated.

TACS employs a high-level centralized controller and a low-level centralized controller to manage and control the thermal status of the cyber components at different levels.

DE is a common algorithm to solve optimal scheduling. This paper is a modified algorithm based on DE.

As shown in Figure 7, this work compares the total number of hotspots and the total hotspot elimination time with three typical methods. It can be seen that this work achieves zero loads to ensure data center stability under all four different experimental load environment conditions. In the fourth scenario, the comparison method, TACS, also achieves zero hotspots, but otherwise does not guarantee complete hotspot elimination. It can be seen that this method takes zero time to eliminate the hotspot, i.e., the cooling energy used to eliminate the hotspot is zero.

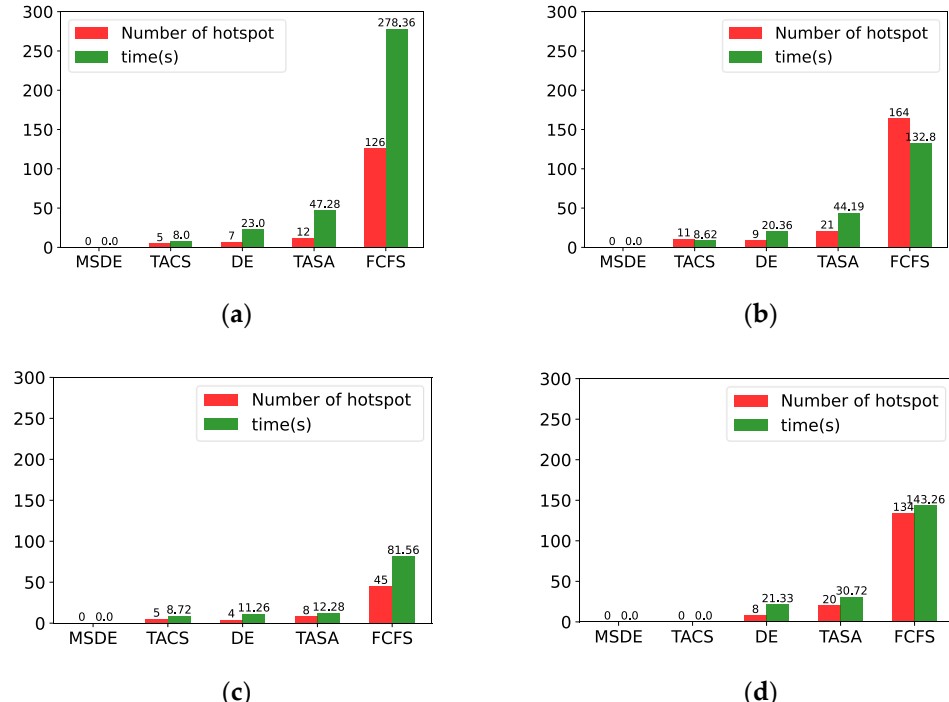

**Figure 7.** Comparison of the number of hotspots and hotspot elimination time results: (**a**) case 1, (**b**) case 2, (**c**) case 3, (**d**) case 4.

In order to better compare the energy consumption between the methods, the results of the runs were visualized for the four different load cases, as shown in Figure 8. As we can see, blue and purple represent energy consumption due to computing and cooling, while yellow and orange represent penalties due to poor system performance. When compared according to the evaluation method proposed in the comparison method text, the average cost saving of this work compared with the comparison method is 378 kWh, 883 kWh, 462 kWh, and 233 kWh for each of the four load conditions, with an average cost saving percentage of 7.96%, 11.1%, 11.3%, and 12.9%, respectively. We can see that the energy cost of our method is slightly lower than other methods, but the penalty cost is extremely low so the overall cost is the lowest. Even evaluated according to the costing method proposed in this paper, this work can have an average cost saving of 2000 kWh for each of the four load conditions.

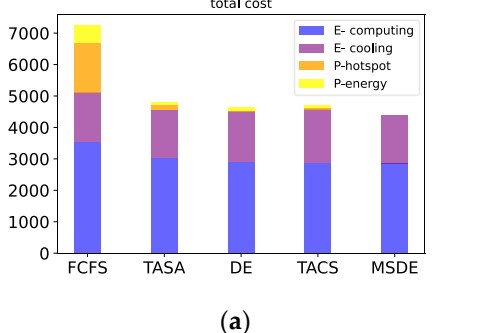

(**a**)

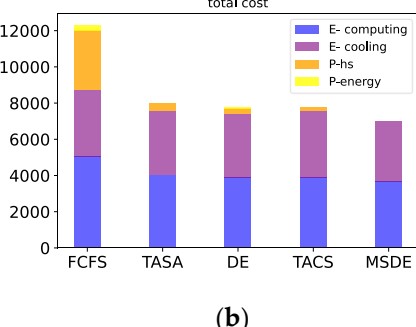

(**b**)

**Figure 8.** *Cont.*

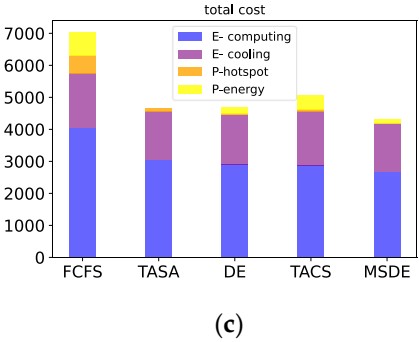 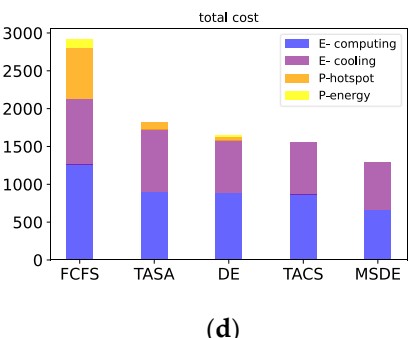

(**c**)                                           (**d**)

**Figure 8.** Comparison with the total cost of the four cases: (**a**) case 1, (**b**) case 2, (**c**) case 3, (**d**) case 4.

Figure 9 shows the heat mapping of the average temperature of each server over a week at the time of input for task 1. As can be seen from the figure, the power control method proposed in this paper controls the temperature in data centers within the safety threshold and achieves the goal of zero hot spots. The specific results are as follows:

(1) Our approach can achieve an average temperature below 25 degrees, ensuring that there is no possibility of hot spots throughout the operation of the data center;

(2) Our approach allows for more uniform heat distribution in data centers than others;

(3) Our approach has the smallest difference between the maximum and minimum temperature of the data center racks, contributing to energy savings.

Compared with the other three methods, it performs best in hotspot elimination for data centers.

In summary, it appears that the power control method proposed in this paper can achieve a zero hot spot response under various load conditions, while at the same time saving a certain amount of energy consumption.

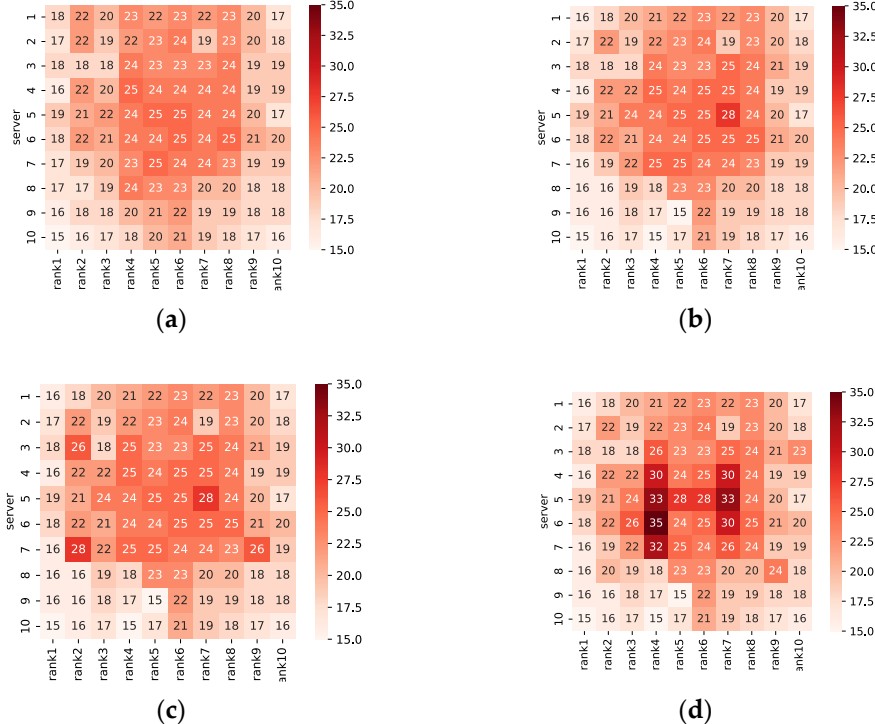

**Figure 9.** *Cont.*

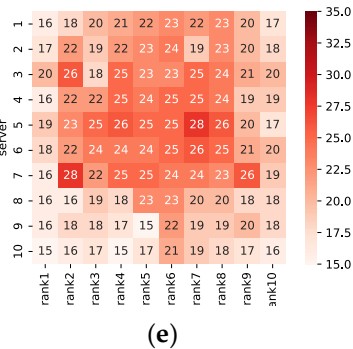

(**e**)

**Figure 9.** Average temperature comparison map: (**a**) MSDE, (**b**) TACS, (**c**) TASA, (**d**) FCFS, (**e**) DE.

## 7. Conclusions

The sustainability and stability of a data center are closely linked to its energy consumption and the number of hotspots. Therefore, it is a great challenge to eliminate hotspots and save energy at the same time. In this paper, a novel power control approach is proposed to achieve energy savings and hotspot elimination in a data center. Specifically, an optimization problem is proposed and solved using an improved differential algorithm, and the results were used for power control. Simulation results based on real data show that this method achieves zero hotspots and lower energy consumption in all applications compared with several existing control methods. In the future, we will further extend our work to further improve the performance of the data center by considering local dynamic changes.

**Author Contributions:** D.L. is responsible for the theme control, theoretical analysis, and experimental scheme design; Y.Z. is responsible for the later revision. J.S., H.L. and J.J. review the paper. All authors have read and agreed to the published version of the manuscript.

**Funding:** This paper is supported by the research Grant from the National Natural Science Foundation of China (Grant No. 62162050).

**Institutional Review Board Statement:** Not applicable.

**Data Availability Statement:** All data supporting the findings in this study are available from the corresponding author on reasonable request.

**Conflicts of Interest:** The authors declare that there is no conflict of interest regarding the publication of this paper.

## Abbreviations

| Symbol | Definition | unit |
| --- | --- | --- |
| $P_{n,i}$ | The power of the $n$-th server in the $i$-th rack at time $t$ | W |
| $P_{idel}$ | The power when the server is idle | W |
| $P_{active}$ | The power when the server is working | W |
| $P_{hs}$ | The total penalty for hot spot presence | ¥ |
| $P_e$ | The penalty for energy overload | ¥ |
| $P_{AC}$ | The power of cooling | W |
| $P_{hn}$ | The number penalty for hot spot presence | ¥ |
| $P_{ht}$ | The time penalty for hot spot presence | ¥ |
| $\overline{P}$ | The nominal parameter for average of power | W |
| $P_{max}$ | The maximum power that can be provided by the data center power supply | W |
| $P_{total}$ | The total energy consumption of all running servers | W |
| $P_d$ | The upper bound of total power demand | W |
| $Q_{in}^i$ | The heat entering the rack | kJ |

| $Q_{out}^i$ | The heat output from the rack | kJ |
| $Q_{rem}$ | The heat that the CRAC has to remove from the air | kJ |
| $Q_{ret}$ | The heat returned from all the racks to the CRAC | kJ |
| $Q_{sup}^i$ | The heat supplied to rack $i$ by CRAC | kJ |
| $T_{sup}$ | The temperature supplied by CRAC | °C |
| $T_0$ | The limit temperature in data centers | °C |
| $C$ | The total cost of a data center | ¥ |
| $C_{computing}$ | The energy consumption of computing | ¥ |
| $C_{cooling}$ | The energy consumption of cooling | ¥ |
| $a$ | The penalty parameter for the number of hotspots appearance | ¥/time |
| $b$ | The penalty parameter for the time of hotspots presence | ¥/s |
| $e$ | The penalty parameter for energy overload | ¥/W |
| $n$ | The number of the working servers | unit |
| $t_i$ | The time when hot spots appear | s |
| $t_j$ | The time when hot spots disappear | s |
| $c_p$ | The specific heat capacity of the air | J/kg·°C |
| $\gamma_{ji}$ | The percentage of flow from rack $j$ to rack $i$ | % |
| $f_i$ | The air flow rate in the rank $i$ | m/s |
| $F$ | Zoom factor | |
| $CR$ | Crossover probability | |

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
