# Peer review of "Energy Saving with Zero Hot Spots: A Novel Power Control Approach for Sustainable and Stable Data Centers"

_sustainability, doi:10.3390/su14159005_

Round 1
Reviewer 1 Report
Comments to the Authors
sustainability-1788357: "Energy-saving with Zero Hot Spots: A Novel Power Control 2 Approach for Sustainable and Stable Data Center", Li Danyang , et al.
The authors have proposed a novel power control approach for energy saving with zero hot spots in data centers.
The review paper will be improved if the following problems are solved.
1. Grammar
It’s better to use “it’s preferred ….” Instead of “we prefer…” throughout the manuscript.
2. Method
As COP is a basic concept, it should explain how Equation 9 is determined.
3. Conclusion
Figure 9 has demonstrated that the proposed method of MSDE works quite efficiently. It’s suggested to summarize 2-4 short items to reveal the key findings.
Author Response
- Grammar: It’s better to use “it’s preferred ….” Instead of “we prefer…” throughout the manuscript.
We corrected this grammatical error and checked and corrected others in the paper.
- Method: As COP is a basic concept, it should explain how Equation 9 is determined.
We have further explained COP and added a relevant reference to aid understanding.
- Conclusion: Figure 9 has demonstrated that the proposed method of MSDE works quite efficiently. It’s suggested to summarize 2-4 short items to reveal the key findings.
We have added a detailed explanation of this, see the paper for details.
Reviewer 2 Report
Please add some recently published works such as:
Mittal, S., 2014. Power management techniques for data centers: A survey. arXiv preprint arXiv:1404.6681. Rong, H., Zhang, H., Xiao, S., Li, C. and Hu, C., 2016. Optimizing energy consumption for data centers. Renewable and Sustainable Energy Reviews, 58, pp.674-691. Arianyan, E., Taheri, H. and Sharifian, S., 2015. Novel energy and SLA efficient resource management heuristics for consolidation of virtual machines in cloud data centers. Computers & Electrical Engineering, 47, pp.222-240. etc. The novelties of the present work have not been stated. Please add the manuscript contributions to the end of the introduction section. The quality of figures should be improved.
Author Response
Please add some recently published works such as:
Mittal, S., 2014. Power management techniques for data centers: A survey. arXiv preprint arXiv:1404.6681. Rong, H., Zhang, H., Xiao, S., Li, C. and Hu, C., 2016. Optimizing energy consumption for data centers. Renewable and Sustainable Energy Reviews, 58, pp.674-691. Arianyan, E., Taheri, H. and Sharifian, S., 2015. Novel energy and SLA efficient resource management heuristics for consolidation of virtual machines in cloud data centers. Computers & Electrical Engineering, 47, pp.222-240. etc. The novelties of the present work have not been stated. Please add the manuscript contributions to the end of the introduction section. The quality of figures should be improved.
We have carefully read, researched and analyzed these papers and cited them.
Reviewer 3 Report
ln 127: See the examples in section 2.3?
ln 167: "....change in air temperature. We get a.."- ".....change in air temperature we get a..."
ln 184: ranks to racks
ln 224-225: penalty parameters a, b. More infromations about the parameters selection, value range, etc.
ln 302: "..n a new parent vector. ????? ,? ∈ {1,2,3,4},..." -"...n a new parent vector. R???? ,? ∈ {1,2,3,4},.."
Author Response
- ln 127: See the examples in section 2.3?
We have modified “section 2.3”to “section 4.2”. This error is caused by a previous modification.
- ln 167: "....change in air temperature. We get a.."- ".....change in air temperature we get a..."
We corrected this grammatical error and checked and corrected others in the paper.
- ln 184: ranks to racks
We corrected this grammatical error and checked and corrected others in the paper.
- ln 224-225: penalty parameters a, b. More infromations about the parameters selection, value range, etc.
We have added a detailed explanation of this, see the paper for details please.
- ln 302: "..n a new parent vector. ????? ,? ∈ {1,2,3,4},..." -"...n a new parent vector. R???? ,? ∈ {1,2,3,4},.."
We corrected this grammatical error and checked and corrected others in the paper.
Reviewer 4 Report
This paper proposes a power control approach to achieve energy savings and hotspot elimination in a data center. It proposes and solves an optimization problem for power control. However, major revision is needed to improve the quality of this paper. The weaknesses are identified as follows:
(1) The Introduction is not well- organized.
(2) Figure1 is unclear and lacks illustration.
(3) In figure 6, the cooling power consumption of case 1 is not that match the tendency of computing power consumption. Why is that? The reason needs to be explained.
(4) In the section of Comparison Results, it’s better to clarify that DE is the variant of MSDE. Only mentioning this in introduction is not appropriate.
(5) Figure 7 is missing the units of number of hotspots and the hotspot elimination time.
(6) The result analysis of figure8 is very unclear.
(7) The authors fail to properly cite several works (e.g., [1-3]) highly related to this work, and clearly discuss the differences between them and this paper.
[1] A Case for Pricing Bandwidth: Sharing Datacenter Networks With Cost Dominant Fairness, TPDS 2021.
[2] MIRAS: Model-based Reinforcement Learning for Microservice Resource Allocation over Scientific Workflows, ICDCS 2019.
[3] Outsourcing Large-Scale Systems of Linear Matrix Equations in Cloud Computing. ICPADS 2016.
(8) The biggest problem is that this article is not well- organized. For example, in the section “Comparison of the total cost of the four cases”, if I want to know what does “p-hs” mean, I need to review equation (10), which is not reader friendly.
Overall, major revision is needed to improve the quality of this paper.
Author Response
- The Introduction is not well- organized.
We have reorganized the Introduction. First, we present the background of the paper and the problems to be solved. Then, We present relevant domestic and international work. Next, we pointed out the shortcomings of the existing work. After this, we present the contribution of our approach. Finally, we have added a section that describes the structure of the paper.
- Figure1 is unclear and lacks illustration.
We switched the position of Figure 1 and Figure 2 to match the description to the image content, which made the message of the figure clear and easy to understand. And we have illustrated the original Figure 1, which is now also Figure 2.
- In figure 6, the cooling power consumption of case 1 is not that match the tendency of computing power consumption. Why is that? The reason needs to be explained.
We have analyzed the reasons for this situation in figure 6. Since there is not always a cooling demand at lower task volumes because less heat is generated by the calculation, there is a mismatch between the cooling and calculation curves.
- In the section of Comparison Results, it’s better to clarify that DE is the variant of MSDE. Only mentioning this in introduction is not appropriate.
We have further illustrated DE and MSDE, see the revised paper for details please.
- Figure 7 is missing the units of number of hotspots and the hotspot elimination time.
We have added them, see the revised paper for details please.
- The result analysis of figure8 is very unclear.
We have further analyzed and illustrated Figure 8. We have specified the meaning expressed in the diagram, for example, what information can be indicated by the color of that segment. Then, we illustrate the advantages of the methods in this paper as compared to their.
- The authors fail to properly cite several works (e.g., [1-3]) highly related to this work, and clearly discuss the differences between them and this paper.
[1] A Case for Pricing Bandwidth: Sharing Datacenter Networks With Cost Dominant Fairness, TPDS 2021.
[2] MIRAS: Model-based Reinforcement Learning for Microservice Resource Allocation over Scientific Workflows, ICDCS 2019.
[3] Outsourcing Large-Scale Systems of Linear Matrix Equations in Cloud Computing. ICPADS 2016.
We have carefully read, researched and analyzed these papers and cited them.
- The biggest problem is that this article is not well- organized. For example, in the section “Comparison of the total cost of the four cases”, if I want to know what does “p-hs” mean, I need to review equation (10), which is not reader friendly.
We have corrected the spelling, grammar and some expressions of thispaper in its entirety. For example, we have modified “p-hs”to “p-hotspot”to enhance the readability of the paper.
Round 2
Reviewer 1 Report
The manuscript is suitable for publication after current revision.
Author Response
Thank you for your review
Reviewer 2 Report
Final decision: accept
Author Response
Thank you for your review
Reviewer 4 Report
The authors have addressed my comments to the previous version. I do not have any further comments, and recommend that this paper be accepted.
Author Response
Thank you for your review